# How can digitalization be used to develop community resilience in public health emergencies?: A qualitative comparative analysis from China

Liqing Li[1], Zihan Li[1]*, Haifeng Ding[1], Meng Gao[1,2]

1 Public Administration and Law School, Hunan Agricultural University, Changsha, China, 2 Economics and Management School, Zhejiang Ocean University, Zhoushan, China

* ZH09537@163.com

**Data Availability Statement:** All indicator data files can be obtained from the China Statistical Yearbook database The repository name for each dataset and the direct link to access each database

## Abstract

Community resilience is critical for the government's response to public health emergencies. With the rapid development of digital technology, leveraging digital tools for grassroots community governance has become increasingly important for the Chinese government. Fuzzy set qualitative comparative analysis (fsQCA) is utilized in this study to establish a framework for investigating the historical development of holistic intelligent governance for community resilience in the context of public health emergencies. Using 31 provincial-level regions in mainland China as research samples and taking the development of community resilience as the outcome variable, the study explores the configurational models and developmental pathways of holistic intelligent governance in enhancing community resilience from a conditional configurational perspective. The results of this study suggest that the mechanisms of community resilience in the face of public health emergencies revolve around five key factors: community self-organization legitimacy, grassroots party-building leadership, policy support, digital governance platform funding, and organizational digitization. However, no single variable alone constitutes a necessary condition for fostering community resilience. Instead, a combination of these factors is required, along with other variables promoting the formation of community resilience, among which community self-organizational legitimacy, grassroots party-building leadership, digital governance platform funding, and organizational digitization serve as the productive basis for promoting the formation of community resilience in holistic intelligent governance at the grassroots level, and are complementary to policy support.

## Introduction

President Xi Jinping emphasized that the community plays a crucial role in the joint prevention and control of epidemics, as well as in defense against external importation and internal spread. The construction of resilient cities, mentioned in the Proposal of the Central

: 1."2021 China Digital Government Construction White Paper" (http://www.caict.ac.cn/kxyj/qwfb/bps/202402/t20240204_471663.htm); 2."Case Collection of Community Creation in China (First Issue)" (https://www.mohurd.gov.cn/gongkai/zhengce/zhengcefilelib/202402/20240202_776557.html); 3."2021 Digital Government Development Index Report" (http://www.caict.ac.cn/kxyj/qwfb/bps/202104/t20210423_374626.htm).

**Funding:** This article is sponsored by the National Natural Science Foundation of China's project "Research on the Formation Mechanism and Improvement Path of Emergency Capacity for Major Public Health Emergencies in Rural Areas" (No. 72274059). Professor Li Liqing, the sponsor, played an important role in the decision to publish the research design.

**Competing interests:** The authors have declared that no competing interests exist.

Committee of the Communist Party of China on the 14th Five-Year Plan for National Economic and Social Development and the 2035 Visionary Goals (hereinafter referred to as "the proposal"), refers to the ability of cities to withstand disasters, mitigate damage, and rationally deploy resources to recover quickly from them [1]. As communities are the fundamental units of society, strengthening community resilience is essential for developing resilient cities and represents a pivotal point in improving national emergency management [2]. The rise of public health events, the normalization of new epidemics, and frequent disruptive occurrences highlight the volatility, complexity, and uncertainty of the social environment. In March 2022, a new wave of the COVID-19 epidemic broke out in Shanghai. In response, grassroots communities played a crucial role in this epidemic, highlighting the importance of community resilience in mitigating and even relieving public health emergencies.

The digital economy is transforming the way grassroots communities conduct business, making grassroots management more efficient and smarter. In particular, throughout the recent epidemic, many communities have embraced digital management as a "learning tool" to improve efficiency, increase total synergy, and optimize resource allocation when faced with public health emergencies. The Recommendations of the Fifth Plenary Session of the 19th Central Committee of the CPC emphasized the need for "digitization to promote innovation in urban and rural development and governance models" and encouraged all sectors of society to participate in "Internet + public services". The modernization of grassroots governance capacity is fundamental to the broader modernization of the national governance system and its operational capabilities. Today, as digital governance continues to advance and integrate into communities, it is essential to test whether existing digital transformation logic is still applicable in the face of sudden public health events and whether digitalization is necessary for community recovery and rebound.

From an emergency management perspective, community resilience is an essential component of a community's emergency management framework. Communities serve as the foundation for effective prevention, control, and response to public health emergencies and act as the "nerve endings" of the national emergency management system [3]. By utilizing digitalization to promote the development of community resilience, it is possible to effectively prevent and respond to risks, maintain the structural stability and normal functioning of the community, and restore its original structure and function quickly after a shock [4]. The use of digitalization to promote the formation of community resilience can effectively prevent and respond to risks, maintain the stability and normal operation of community structures and functions, and and enable a quick recovery of these elements after a shock [5]. Therefore, there is a high degree of fit and commonality between building community resilience and improving the emergency management system, and utilizing digital technology to create a resilient community public health emergency management system offers a new perspective for communities to effectively respond to public health emergencies.

Nowadays, the modern concept of community resilience is rooted in the intersection of ecology and psychology. In the field of ecology, resilience focuses on the study of system dynamics and describes the continuity and stability of external systems in the face of major changes, providing a rich analytical concept of community resilience [6]. In the field of psychology, resilience highlights how communities adapt to adversity. This concept is also embedded in grassroots community governance, which fosters the development of "community resilience" as communities manage and respond to dramatic shifts in their survival environment compared to earlier times [7]. When studying community resilience, we discovered that it spans multiple domains, including economy, society, engineering, and ecology. The concept of community resilience has found widespread application in disaster response, particularly in the context of the frequent global disaster events of recent years. For instance, the wildfires in

Australia, the super typhoon Lekima, and the deadly heatwaves sweeping across much of Europe continue to pose challenges to the resilience of communities. Despite being infrequent, public health emergencies can spread swiftly and pose a severe risk to human health. In recent years, outbreaks such as SARS, Ebola, influenza, and COVID-19 have had a profound impact on people's life safety and the normal operation of society. Therefore, against this backdrop, the significance of community resilience building has been elevated to a new level.

Community resilience refers to a community's ability of a community to learn from each crisis, restore its original structure and functions, and be highly capable of learning and adapting. At this stage, due to varying research focuses, scholars have differing interpretations and applications of the concept, which are mainly divided into two categories: "process-oriented" and "capacity-oriented". The "process-oriented" approach views community resilience as a positive developmental process in which communities demonstrate functionality and adaptability after encountering external disturbances [8], while the "capacity-oriented" approach views community resilience as a community's ability to recover from unexpected situations, including different forms of coping, stabilization, recovery, and adaptation. This approach considers community resilience as the ability of communities to cope with emergencies, including different forms of capacity such as coping, stabilization, recovery, and adaptation. It sees community resilience as a product of the interaction between organizational capacity and the environment, typically defining community capacity as the mobilization of social resources and the ability to utilize them to address emergencies at the grassroots level [9].

Some scholars support that the formation of community resilience requires the community to complement the top-down public health emergency management system through bottom-up linkage, which in turn strengthens the foundation of the urban emergency management system [10]. Various models, such as the community disaster resilience framework [11], Luo Qiangqiang's community resilience model [12], and the community resilience model for community public health emergencies, have been proposed. These models correlate the elements of self-organization, social capital, and intelligent governance, which provide a reference for improving community resilience. Additionally, some scholars suggest moving from a vertical, fragmented, and compartmentalized approach to horizontal, synergistic, and holistic community resilience construction. This approach forms a pluralistic and synergistic practice model, where the Party organization leads and encourages the participation of all sectors of society [13].

Holistic intelligent governance, a tool that combines the concept of holistic governance with digital governance, can help address challenges in grassroots management through institutional regulatory methods [14]. It primarily relies on the adjustment of vertical and horizontal relationships to solve governance dilemmas, ensuring that grassroots are not left with responsibilities that exceed their available resources or authority [15]. However, holistic governance itself remains constrained by traditional management thinking, requiring constant "empowerment" from higher levels " to drive effective governance [16]. With the rise of digital technologies such as artificial intelligence, cloud computing, and big data, digital governance has become an important opportunity for grassroots communities to improve their means of governance and enhance their effectiveness. However, it also faces challenges such as collaborative dilemmas and information silos, technological rationality, institutional lag, technological risks, and governance crises [17,18]. In practice, issues such as difficulties in data integration, poor quality of technical tools and training, ineffective outreach in rural areas, and the overall poor level of digital governance performance of grassroots communities restrict the modernization process of grassroots governance [19]. Grassroots digital governance is an inevitable result of the modernization of governance capacity, and its transformation and upgrading can help optimize operational processes and improve efficiency [20]. Using intelligent thinking,

ways, and means to reconstruct grassroots governance structures, reorganize governance resources, reshape governance functions, and form new governance structures and governance mechanisms can provide new and digital solution ideas for comprehensively promoting the modernization of grassroots governance in all aspects and dimensions [21].

After reviewing the above literature, it is evident that there are few studies focused on using "holistic intelligent governance" to promote community resilience. Existing studies mostly focus on singular aspects such as technology, environment, capital, or management, and pay little attention to the construction of "community resilience". Additionally, the focus on "community resilience" is often overshadowed by the focus on "resilient cities". Zhang et al. [22], Li et al. [23], and Ma et al. [24] studied "residents' preparedness", "economic inequality", "wealth disparity", "neighborhood relationship", and "social norms" regarding community resilience from the perspectives of technology and environment. From the capital perspective, Manzi [25] and Chandra et al. [26] analyzed the impact of social capital on community resilience. From the perspective of emergency management, Lan et al. [27] argued that the improvement of community resilience should be integrated into the process of grassroots governance innovation and governance system modernization from the perspective of emergency management.

This article, grounded in grassroots communities in China, examines how holistic intelligent governance enhances community resilience during sudden public health emergencies. Utilizing the complex dynamic perspective of configuration theory, it integrates both qualitative research and measurement-based approaches to innovatively introduce a theoretical analysis framework of "holistic intelligent governance—community resilience formation". Employing the fsQCA method, it conducts a case analysis on data samples from 31 provincial-level regions in mainland China. By investigating the conditional configurations and pathways through which holistic intelligent governance fosters community resilience in response to sudden public health emergencies, the article elucidates how holistic intelligent governance promotes community resilience during such emergencies. This provides a theoretical and evidence-based foundation for enhancing communities' emergency management capabilities and shaping community resilience research.

## Data and methods

### Variable design

In this paper, we review and synthesize existing literature on the construction of community resilience models in the context of public health emergencies. We draw from the works of several scholars, including Shi's community disaster resilience framework and Xu's "state-society" interaction framework, which explore the correlation between community party organizations, community social capital, policy support, and other factors in building community resilience. Additionally, we discuss the role of digital technology in promoting community resilience, referencing the works of Luo's community resilience model and Cutter's community resilience models, which propose the use of data empowerment technology, grassroots community systems, and community self-organization autonomy. Building on Li's proposed emergency response capacity generation mechanism [28], we integrate holistic governance theory and digital governance theory to develop a theoretical analysis framework for "holistic intelligent governance-community resilience formation". We then construct a model of holistic intelligent governance group linkage that considers the legitimacy of community self-organization, the leadership of grassroots party building, policy support, funding of digital governance platform, and digitization of organizational institutions, and the mechanism of holistic intelligent governance group linkage influencing community resilience formation in

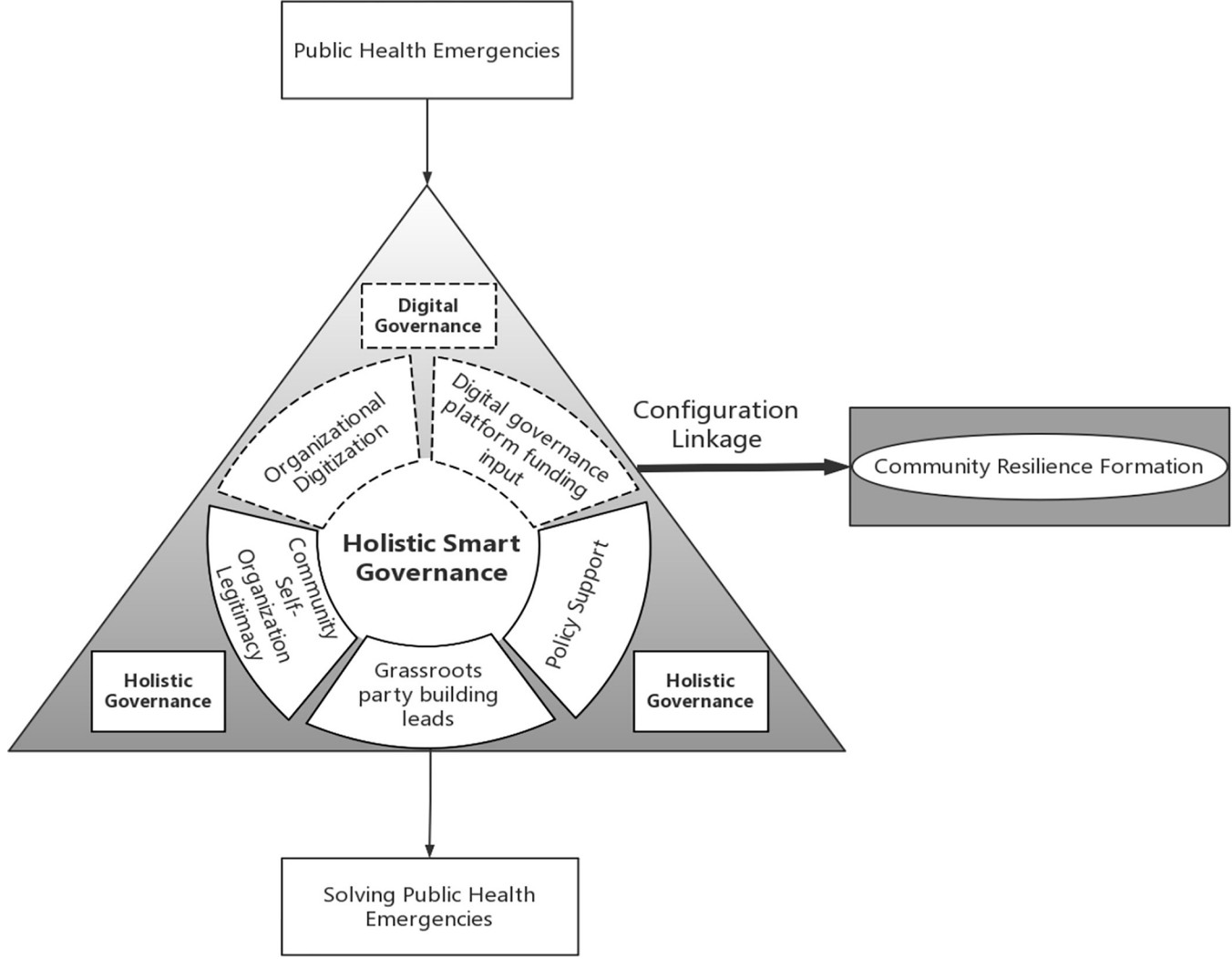

**Fig 1. The theoretical analysis framework of "holistic intelligent governance—community resilience formation" group linkage.**

the face of public health emergencies. The mechanism behind resilience formation (as illustrated in Fig 1) and the definitions of various variables in the analysis are as follows.

## Condition variables

①The legitimacy of community self-organization can be divided into normative legitimacy and cognitive legitimacy, as discussed by Xu et al. [29]. Normative legitimacy refers to whether the community self-organization adheres to corresponding rules and systems, while cognitive legitimacy concerns recognition from the government and the public. When the government encourages the formation and operation of community self-organizations and provides financial support to community self-organizations at the beginning of their inception, it can be seen as recognition of their legitimacy. To gather data on this variable, we utilized the Case Collection of Community Creation in China (First Issue) and the research summaries from scholars such as Xu [30].

②Leadership of grassroots party building. As noted by Wang [31], communities often rely heavily on grassroots leaders in the face of public health emergencies. In particular, conducting

timely party-building activities and enhancing the digital literacy of grassroots leaders are key to ensuring an effective response. Building on existing research, we evaluate whether monthly party-building activities are consistently conducted on time and whether digital literacy training sessions for grassroots leaders have been held to assess the overall effectiveness of grassroots party-building. If the sample indicates any of the following situations: the actual leader of the organization serves as the secretary of the party branch, or party-building activities are conducted on time every month, assign a value of 1; otherwise, assign a value of 0. The relevant data were collected from the government websites and the research summaries by scholars such as Wang Xuejun and other scholars.

③Policy support. Drawing on the ideas of Zhao Yan et al. [32], we investigate to what extent the release of higher-level policies can stimulate or constrain grassroots adoption behavior when building "resilient communities" in various regions. This is measured by the number of policies issued by each region that mention the construction of resilient communities. In the context of China, the more policies there are, the more emphasis they place on this work, and the more favorable it is for the promotion of this work. This article draws on existing research and assigns a value of 1 to regions that have specifically formulated a resilient community construction plan or a resilient city construction plan; otherwise it is assigned a value of 0.

④ Funding input for digital governance platform. As Li et al. emphasize [33], investing in digital governance platforms is crucial for building intelligent communities and enhancing community resilience [34]. In this study, we measure the funding input for digital governance platforms by assessing the level of digital development in each region, using data from the Digital Government Development Index Report (2021) based on the "Digital Economy Development Level" score. Measuring the "Digital Economy Development Level" is a complex and multidimensional process, encompassing various evaluation indicators. Some key measurement dimensions and specific indicators include: digital economy infrastructure construction, digital economy industry scale and growth, digital economy innovation capability, digital economy application and integration, digital economy governance and regulation, as well as the social impact of the digital economy. In China, the assessment of the development level of the digital economy typically relies on authoritative data sources, such as statistical data released by government departments like the National Bureau of Statistics and the Ministry of Industry and Information Technology of China.

⑤ Digitization of organizations. We draw upon the perspective of Lan et al. [27], who argue that enhancing the construction of digital government is crucial for promoting efficient and effective intelligent governance in communities. The degree of digitization in organizations is measured through the digitalization of party and government institutions, as detailed in the "2021 China Digital Government Construction White Paper". This article evaluates party and government organs across various provincial administrative divisions in China, based on the evaluation criteria outlined in the "2021 China Digital Government Construction White Paper". These criteria encompass a range of comprehensive indicators, including the prevalence of government cloud adoption, the bandwidth capacity of government networks, the number and processing capabilities of data centers, the number of departments connected to data-sharing platforms, the rate of data openness, the frequency of cybersecurity incidents, the efficiency in addressing data security vulnerabilities, and the speed of activating emergency response mechanisms.

**Outcome variables.** In this paper, the outcome variable is the development status of community resilience, which is constructed based on the work of Shi et al. [11]. We selected five indicators [35,36] to assess the development status of community resilience, which are based on the community's digital governance attributes. These indicators include community

Table 1. Basic information of variables.

| Variable Type | Variables | Indicators | Data Source |
|---|---|---|---|
| **Condition Variables** | Community Self-Organization Legitimacy | Government Public Recognition level | Case Collection of Community Creation in China (First Issue) |
| | Grassroots Party Building Leadership | The Degree of Perfection of Grass-Roots Party Building | Government Websites And News Reports |
| | Policy Support | Leadership Importance | Government Official Websites and Other Policy Resource Platforms |
| | Digital Governance Platform Funding | Digital economy Development Level | Digital Government Development Index Report (2021) |
| | Organizational Digitization | The degree of construction of Digital Government | White Paper on Building Digital Government in China 2021 |
| **Outcome Variables** | Community Resilience Development Status | Community Resilience Development Status | Literature Analysis |

organizational resilience, grassroots digital governance development, social resilience, grassroots economic resilience, and grassroots digital facilities. The literature on community resilience research is used to guide the selection of these indicators.

## Data sources

In this paper, we use data from 31 provincial-level regions in mainland China are selected as the sample. The original data for the independent and dependent variables are mainly obtained from various sources, including the "2021 China Digital Government Construction White Paper", "Case Collection of Community Creation in China (First Issue)", "2021 Digital Government Development Index Report", and "National Digital Economy Development Index (2021)". Data on grassroots party building, policy support, and other government affairs primarily come from national policy information resource network platforms, such as the portals of provincial people's governments. To ensure the comprehensiveness and completeness of the data, this paper defines the collection scope of policy measures related to communities issued by provincial and municipal governments in each region during public health emergencies from a broad perspective. Table 1 presents the basic information of all variables is shown in Table 1.

## Method

Qualitative Comparative Analysis (QCA) is a research method that was developed by sociologist Ragin in the 1980s. QCA is based on the idea that complex social phenomena can be understood by examining the links between different groupings and identifying the complex causal relationships between the elements of these groupings and the expected outcomes. This method aims to answer questions such as "What are the combinations of elements that lead to the desired outcome?" and "What is the core element in a given combination of elements?"

In this paper, we use the fsQCA method to study the factors that influence the formation of community resilience in the face of public health emergencies in 31 provincial-level regions in mainland China. In fsQCA, a variable is assigned a value ranging from 0 to 1 according to its difference from the ideal value [37], providing a more precise reflection of the variable's true state. Since different approaches to assigning values to variables can lead to varying outcomes, we employ fsQCA to fulfill the specific needs of our study. This approach helps us to identify a multifaceted path for enhancing emergency management capacity, promoting holistic intelligent community governance, and fostering community resilience in response to public health emergencies, while also exploring the relationships between different group states.

**Table 2. Variable calibration.**

| Variable Name | | Positioning Points | | |
| --- | --- | --- | --- | --- |
| | | Completely Unaffiliated | Middle Point | Completely Affiliated |
| **Condition Variables** | Community Self-Organization Legitimacy | 0.805 | 0.930 | 1.220 |
| | Grassroots Party Building Leadership | 33.350 | 50.900 | 74.300 |
| | Policy Support | 5.500 | 23.000 | 42.500 |
| | Digital Governance Platform Capital Investment | 12.625 | 45.400 | 100.165 |
| | Organizational Digitization | 4.500 | 8.000 | 10.000 |
| **Outcome Variable** | Community Resilience Development Status | 7.850 | 10.501 | 18.100 |

## Results

### Data calibration

The fsQCA method employs set theory to convert the conditional and outcome variables into sets ranging from [0–1]. In this study, the set of 31 provincial-level regions in mainland China is used to represent a particular element as a fuzzy set membership score. To conform to the Boolean logic of the QCA method, the original fuzzy set data is calibrated and converted into a truth table, thereby combining the advantages of both qualitative research and study on measurement. Three qualitative anchor points are used for calibration in this paper, namely "no affiliation at all (5%)", "midpoint (50%)" and "completely affiliated (95%)", based on the actual meaning of the data and vector attributes. The calibration information of the sample data is presented in Table 2.

### Truth table construction

In this paper, we adopted the method of triangulation verification to enhance the reliability of the data. This involves using multiple sources of information to cross-validate the data and construct the truth table per the coding requirements, which are shown in Table 3.

### Necessity analysis of a single conditional variable

The method of necessity analysis method is used to examine if a single variable has a direct impact on the outcome, by setting a minimum threshold of 90%, based on the principles of consistency and coverage. If the consistency is higher than 90%, it indicates that the variable must influence the outcome. In this study, we employed the fsQCA 3.0 software in this study to carry out the analysis, constructing a standardized truth table for the five conditional variables based on conditional configurations. The results are presented in Table 4. This approach allowed us to further investigate the complex causal relationships between holistic intelligent governance and the enhancement of community resilience in the context of public health emergencies.

### Sufficiency analysis of conditional grouping

The fsQCA 3.0 software is employed to analyze the three types of solutions: "complex solution", "intermediate solution", and "simple solution". As this paper prioritizes accepting "logical remainder" for intermediate solutions, which is more consistent with its research focus, the intermediate solutions are selected to analyze the conditional groupings. This approach provides more objective information for analyzing research results and is widely used in many similar research methods.

**Table 3. Truth table.**

| No. | Condition Variables | | | | | Outcome Variable | Corresponding Cases |
|---|---|---|---|---|---|---|---|
| | Community Self-Organization Legitimacy | Grassroots Party Building Leadership | Policy Support | Digital Governance Platform Capital Investment | Organizational Digitization | Community Resilience Development Status | |
| 1 | 1 | 1 | 0 | 0 | 1 | 1 | Sichuan, Chongqing, Guangzhou |
| 2 | 0 | 1 | 0 | 1 | 1 | 1 | Shanghai, Zhejiang |
| 3 | 0 | 1 | 1 | 1 | 1 | 1 | Hubei, Jiangxi, Shandong |
| 4 | 1 | 1 | 1 | 1 | 1 | 1 | Beijing, Tianjin, Jiangsu |
| 5 | 1 | 0 | 1 | 0 | 1 | 0 | Jilin, Heilongjiang |
| 6 | 0 | 0 | 1 | 1 | 0 | 0 | Hunan, Yunnan |
| 7 | 1 | 0 | 0 | 0 | 0 | 0 | Qinghai, Ningxia |
| 8 | 1 | 0 | 1 | 0 | 0 | 0 | Hainan, Inner Mongolia |
| 9 | 0 | 0 | 0 | 0 | 1 | 0 | Liaoning, Guangxi |
| 10 | 1 | 1 | 0 | 0 | 0 | 0 | Fujian, Guizhou |
| 11 | 0 | 1 | 1 | 0 | 0 | 0 | Anhui |
| 12 | 1 | 1 | 0 | 1 | 0 | 0 | Hebei |
| 13 | 0 | 0 | 0 | 1 | 0 | 0 | Gansu |
| 14 | 0 | 1 | 0 | 0 | 1 | 0 | Henan |
| 15 | 1 | 1 | 1 | 0 | 0 | 0 | Shaanxi |
| 16 | 1 | 0 | 1 | 1 | 0 | 0 | Shanxi |
| 17 | 0 | 0 | 1 | 0 | 0 | 0 | Xinjiang, Tibet |

Through the analysis of the five selected conditional variables, a histomorphic configuration was formed, resulting in the identification of four causal combination paths, which are presented in Table 5. The combined coverage of these paths was found to be 0.703636, indicating their ability to explain more than 70% of the cases. Moreover, the overall consistency of the four paths was 0.808164, which exceeds the acceptable reference value of 80%. As such, it can be concluded that these four groupings sufficiently account for the outcome variables in this study.

**Table 4. Univariate necessity test.**

| Condition Variablesz | Digital Intelligence Empowerment for Community Resilience Enhancement | |
|---|---|---|
| | Consistency | Coverage |
| **Community Self-Organization Legitimacy** | 0.671557 | 0.648311 |
| **Non-Community Self-Organization Legitimacy** | 0.699828 | 0.591638 |
| **Grassroots Party Building Leadership** | 0.782422 | 0.710239 |
| **Non-Grassroots Party Building Leadership** | 0.586888 | 0.525372 |
| **Policy Support** | 0.750930 | 0.621490 |
| **Non-Policy Support** | 0.546092 | 0.540445 |
| **Digital Governance Platform Funding Input** | 0.688663 | 0.679472 |
| **Non-Digital Governance Platform Funding Input** | 0.645577 | 0.535661 |
| **Organizational Digitization** | 0.685156 | 0.707696 |
| **Non-Organizational Digitization** | 0.614729 | 0.491558 |

**Table 5. Configuration results.**

| Condition Variables | Configuration Structure | | | |
|---|---|---|---|---|
| | **Path 1** | **Path 2** | **Path 3** | **Path 4** |
| Community Self-Organization Legitimacy | ● | | ⊗ | ● |
| Grassroots Party Building Leading | ● | ● | ● | ● |
| Policy Support | | ⊗ | ● | ● |
| Digital Governance Platform Funding | ⊗ | ● | ● | ● |
| Organizational Digitization | ● | ● | ● | ● |
| Original Coverage | 0.350701 | 0.343544 | 0.380261 | 0.347839 |
| Unique Coverage | 0.101632 | 0.0802318 | 0.119095 | 0.004294 |
| Consistency | 0.816531 | 0.79721 | 0.834983 | 0.788833 |
| Total Coverage | 0.703636 | | | |
| Total Consistency | 0.808164 | | | |

Note: The presence of core variables is indicated by ●, the presence of auxiliary variables is indicated by ●, the absence of variables is indicated by ⊗, and the presence or absence of variables is indicated by "blank". The core variables play a key role. The auxiliary variable corresponds to the core variable, but it has a limited effect on the variation of the results.

Through an analysis of the histotype structure and a comparison of the simple and intermediate solutions, this study reveals that community self-organization legitimacy, grassroots party-building leadership, funding input for digital governance platforms, and organizational digitization play crucial roles in each of the four histotype configurations. These core conditions are integral in utilizing holistic intelligent governance to promote community resilience when communities face public health emergencies. The key condition in driving community resilience through holistic intelligent governance is further emphasized. Additionally, considering China's current state of China's digital government construction and digital ecological development, this study identifies the four core conditions as community self-organization legitimacy, grassroots party-building leadership, digital governance platform funding, and digitization of organizational structure. The four paths of holistic intelligent governance to promote community resilience are grouped into three models, with paths 2 and 3 sharing the core conditions of funding for digital governance platforms and organizational digitization. The other conditions are either missing or have a limited effect on the variability of the results, resulting in similar development patterns. Therefore, the two paths are combined into one mode.

Condition grouping 1, labeled as "Holistic governance-driven" in Table 5, suggests that the convergence of three core conditions—community self-organization legitimacy, grassroots party leadership, and organizational digitization—can promote community resilience in the face of public health emergencies. This promotion can be realized with relatively low financial investment in digital governance platforms, as long as the community has a strong social foundation, a solid environmental system, and a well-developed digital infrastructure. Representative cases for this grouping include Sichuan Province, Chongqing City, and Guangzhou Province. For instance, Chongqing, a key province in the southwest of China, may not have the same economic, educational, and technological advantages as developed regions, but it is one of the core cities in the Yangtze River Economic Belt that directly radiates the "One Belt, One Road" initiative through its regional resources and industrial base. By taking advantage of its strong regional industrial base, Chongqing can absorb the scientific and financial resources from developed eastern coastal regions such as Jiangsu, Zhejiang, and Shanghai. Moreover, Chongqing, a municipality that blends a large urban center, agriculture, mountainous terrain,

and reservoirs, can enhance the integration of science and technology service industries with community service capacity building, ultimately improving community resilience and response capacity during public health emergencies.

"Driven by Digital Governance" corresponds to Condition Grouping 2 and Condition Grouping 3 in Table 5. Given that the core conditions of both groupings are identical and the remaining conditions either are missing or have a limited effect on the outcome, they are consolidated into a single category.

Condition grouping 2 comprises two core conditions: digital governance platform funding and organizational digitization, supported by the grassroots party leadership. This grouping suggests that communities can enhance their resilience in response to public health emergencies, if they have sufficient investment in digital governance platforms, supported by a well-structured digital infrastructure and grassroots party-building efforts. Representative cases are Shanghai and Zhejiang Province, both of which are the economic, demographic, and technological powerhouses of China. With a concentration of internet-based enterprises, numerous technology innovation platforms, and rich educational and technological resources, these regions have established a high-quality digital intelligence environment for grassroots communities. For instance, Shanghai launched the "Shanghai 78" initiative, a community-based public health emergency response plan that creates a "community resilient living circle" by enhancing the collaborative capacity of grassroots community organizations and their ability to respond to public health emergencies. In Hangzhou, Zhejiang Province has implemented the "digital intelligence empowerment and three-dimensional protection to build a resilient city in urban and rural areas", which proposed the use of digital technology to promote community resilience, especially in sensitive urban areas. These regions have a well-developed digital intelligence infrastructure and a high level of participation in community building. Consequently, with a small investment, the government can empower digital intelligence management to all aspects of community governance during public health emergencies, accelerating the formation of community resilience.

In Condition Grouping 3, the combination of digital governance platform funding investment and organizational digitization are the two core conditions that promote community resilience. When supported by grassroots party-building leadership and policy support, these conditions can further enhance resilience. With adequate funding for digital governance platforms and well-established digital organizations, communities can effectively respond to public health emergencies. Representative cases include Hubei Province, Jiangxi Province, and Shandong Province. Located in the heart of the Yangtze River Economic Belt, Hubei Province's Huangshi City was one of four Chinese cities selected for the global "100 Resilient Cities" initiative. With extensive experience in building community resilience, the province has established a leading group for resilient city development to address public health emergencies. It has also invested in digital governance platforms and enhanced organizational digitization, while integrating smart technologies into community management services to foster a supportive environment for community resilience. With support from grassroots party-building leadership and policy, this strategy has been successful in promoting resilience. By leveraging the technical resources of the East and West, Hubei Province has demonstrated how funding investment in digital governance platforms and organizational digitization can be used to enhance community resilience.

In condition grouping 4, the core conditions of sufficient funding for digital governance platforms and a sound organizational structure being digitally erected are necessary for promoting community resilience. Additionally, community self-organization legitimacy, a more complete grassroots party-building leadership system, and a small amount of policy support can supplement these core conditions. Representative cases of this approach include Beijing,

Tianjin, and Jiangsu Province. The "Holistic Governance + Digital Governance Driven" approach to community resilience emphasizes a holistic perspective. For instance, Beijing issued the "Guidance on Accelerating the Construction of Resilient Cities" in 2021, which mandates the completion of 50 community resilience, neighborhood resilience, and other resilience projects in Beijing by 2025. This initiative aims to create a comprehensive index and standards system that offers valuable insights and experiences for different types of areas, providing a model to learn from and replicate. Beijing is a core city of China located in the Beijing-Tianjin-Hebei region and has unparalleled advantages in terms of economy, education, science, technology, and policy-oriented resources. The city's booming intelligent manufacturing, the Internet, and other new high-tech industries bring together many leading domestic and international enterprises. The government and market "two hands" complement each other to promote the development of the digital governance platform in community resilience building. The technology service industry's technological advantages help to accelerate regional technology exchange and promotion, thus improving the community's ability to respond to public health emergencies and enhancing grassroots community management services. This approach accelerates the formation and improvement of community resilience.

### Robustness tests

To validate the model, we referred to authoritative research findings [37]. The original model was adjusted by changing the "completely unaffiliated" (5%) and "completely affiliated" categories from 5% and 95%, respectively, to 25% and 75%. The "completely affiliated" category was set at 75%. The rest of the model was left unchanged and analyzed again. This adjustment resulted in significant changes in the core conditions and configuration paths, indicating that the grouping results can be considered robust.

## Discussion

### General discussion

In the new era of digital intelligence, governments are increasingly leveraging digital technology to strengthen grassroots community governance, particularly during public health emergencies. This trend reflects the changing nature of digital intelligence technology. However, the main contradiction in society is manifested in the unbalanced and inadequate development at the level of grassroots community construction. To meet the public's demand for "community services + Internet", it is essential to improve the level of "Holistic Governance + Digital Governance". The objective of this study is to promote community resilience formation through holistic intelligent governance in response to public health emergencies. Based on relevant literature, we identify the five conditional variables: community self-organization legitimacy, grassroots party-building leadership, policy support, financial investment in digital governance platforms, and the degree of organizational institutional digitization. Using the fsQCA analysis method, we investigate the conditional grouping and pathways of holistic intelligent governance for community resilience formation in the face of public health emergencies.

This study has demonstrated that promoting community resilience in the face of public health emergencies through holistic intelligent governance is not dependent on a function of a single factor, but rather the result of a combination of factors. Our analysis revealed three main types of governance: "Holistic Governance Driven", characterized by stronger legitimacy of community self-organization and more robust grassroots party-building leadership; "Digital Governance Driven" with greater funding for digital governance platforms and improved digital organizational structures; and "Holistic Governance + Digital Governance Driven" model,

which incorporates a complete digital governance platform, a well-developed digital organization, an extensive public base, a solid grassroots party building system, and policy support. Furthermore, this study found that community self-organization legitimacy, grassroots party-building leadership, funding for digital governance platforms, and digitization of organizational structures are core conditions for promoting community resilience formation through holistic intelligent governance in the face of public health emergencies. While policy support serves as an auxiliary condition for this model, it is not a core requirement, as its importance fluctuates with the level of economic development and the priorities of grassroots work in different regions. As such, region-specific analysis and policy support are necessary. Overall, promoting community resilience through holistic intelligent governance for community resilience requires multiple concurrent and symbiotic elements, which collectively shape the community resilience formation system in the face of public health emergencies.

The theoretical framework for "holistic intelligent governance-community resilience formation" presented in this article centers on the driving factors and transformational paths. It summarizes relevant indicators for resilience community evaluation in both Chinese and Western academic circles. Additionally, a configuration path analysis was conducted on Chinese community resilience cases, summarizing the four paths through which holistic governance promotes community resilience formation in the face of public health emergencies into three models. The models suggest that "holistic governance" and "digital governance" jointly promote community resilience formation. These suggestions provide a theoretical basis and practical direction for local governments to enhance their emergency management capabilities and advance research on community resilience in response to public health emergencies. Taking into account the uneven economic and social development levels in various regions of our country and the differing capacities of local governments, the path to improvement should not be generalized.

The four combination paths identified through fsQCA guide local governments in integrating local realities, adjusting to regional and temporal conditions, and developing a strategic plan for fostering community resilience in a stepwise and region-specific manner. This offers both a theoretical framework and an evidence-based foundation for strengthening communities' emergency management capabilities in response to public health emergencies and advancing community resilience research. It holds significant value for shaping a digital government and furthering the modernization of the national governance system and its capacity. The theoretical analysis framework of "holistic intelligent governance-community resilience formation" constructed in this study can offer new perspectives and ideas for countries and regions with similar national conditions in China in the field of community resilience research. However, when applying this framework, each region still needs to optimize and supplement it according to its own actual situation.

## Policy insights

This paper presents a qualitative comparative analysis of the mechanisms behind community resilience formation in China during public health emergencies. The analysis yields valuable policy insights for local governments seeking to enhance community resilience in the face of such emergencies.

Our findings suggest that the effective promotion of community resilience through holistic intelligent governance to drive community resilience relies on a multi-factor grouping of coupled mechanisms. Promoting community resilience requires a combination of multiple factors, including community self-organizational legitimacy, policy support, grassroots party leadership, funding for digital governance platforms, and digitization of organizations. If any

of these factors are missing, a "barrel effect" may arise, undermining the ability of holistic intelligent governance to promote community resilience.

When confronted with public health emergencies in the community, the government can proactively enhance human, financial, and material resources. This can include conducting comprehensive risk assessments of physical systems and engineering facilities in the community, establishing a risk perception system using intelligent systems and modern technology, and monitoring facility data in real time to quickly identify and prevent safety crises. Such measures will lay a strong foundation for "Holistic Governance + Digital Governance", inject momentum into community resilience, and promote its formation through government-initiated changes that extend to the community level.

However, due to regional differences in development conditions and governance stages, local governments must adopt tailored, region-specific approaches to community resilience. Given that regions vary in their foundational conditions and the types of public health emergencies they encounter, each region must design a "fit-for-purpose" approach, considering digital development levels, human characteristics, and institutional contexts to maximize effectiveness.

Our analysis highlights four core conditions for effective community resilience formation: community self-organization legitimacy, grassroots party-building leadership, funding for digital governance platforms, and organizational digitization. To effectively promote community resilience, governments must adopt a "top-level design" approach, serving as both "helmsman" and "servant" to empower grassroots organizations and promote a systematic, comprehensive, and coordinated manner. The government moved away from being a "paddler" and embraced the roles of "helmsman" and "servant", maintaining an open approach and granting grassroots communities full empowerment, which led to more effective governance.

To achieve these goals, governments must also embrace the digital reconstruction of grassroots community governance, formulate a reasonable strategy for digital intelligence as a means of modernizing governance capacity, and foster incremental growth in community management. By breaking free of conventional contexts and stimulating the "growth force" of modernization, governments can ensure that communities are better equipped to respond to public health emergencies in a timely and effective manner.

## Conclusion

In this paper, we present a histological research framework for the formation of community resilience in the face of public health emergencies, using fuzzy set fsQCA. Our study sample consisted of 31 provincial-level regions in mainland China as the study sample, and we used the community resilience development status as the outcome variable. We analyzed the group configuration and development path of holistic intelligent governance for community resilience formation in the face of public health emergencies from a conditional grouping perspective. Our findings suggest that the mechanism of action of holistic intelligent governance relies on five key factors: community self-organizational legitimacy, grassroots party-building leadership, policy support, funding for digital governance platforms, and digitization of organizational structures. A single variable is not enough to drive the formation of community resilience; instead, it necessitates the integration of multiple variables. In particular, community self-organization legitimacy, grassroots party-building leadership, digital governance platform funding investment, and organizational digitization are the productive basis for promoting community resilience in holistic intelligent governance at the grassroots level and are complementary to policy support. Our research contributes to the understanding of the complex mechanisms underlying the formation of community resilience in the face of public

health emergencies and provides practical recommendations for promoting community resilience through holistic intelligent governance.

## Author Contributions

**Conceptualization:** Liqing Li.

**Data curation:** Zihan Li.

**Formal analysis:** Liqing Li.

**Funding acquisition:** Liqing Li.

**Methodology:** Haifeng Ding, Meng Gao.

**Resources:** Zihan Li.

**Software:** Zihan Li.

**Supervision:** Haifeng Ding, Meng Gao.

**Validation:** Haifeng Ding, Meng Gao.

**Writing – original draft:** Zihan Li.

**Writing – review & editing:** Meng Gao.

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
