## [Decision Letter · Decision Letter 0]

1 Sep 2024

PONE-D-24-10967How does community resilience develop in public health emergencies?: A qualitative comparative analysis from ChinaPLOS ONE

Dear Dr. li,

Thank you for submitting your manuscript to PLOS ONE. After careful consideration, we feel that it has merit but does not fully meet PLOS ONE’s publication criteria as it currently stands. Therefore, we invite you to submit a revised version of the manuscript that addresses the points raised during the review process.

We look forward to receiving your revised manuscript.

Kind regards,

Shenghua Xie

Academic Editor

PLOS ONE

“This article is sponsored by the National Natural Science Foundation of China's project "Research on the Formation Mechanism and Improvement Path of Emergency Capacity for Major Public Health Emergencies in Rural Areas" (No. 72274059).”

3. Please note that your Data Availability Statement is currently missing the repository name and/or the DOI/accession number of each dataset OR a direct link to access each database. If your manuscript is accepted for publication, you will be asked to provide these details on a very short timeline. We therefore suggest that you provide this information now, though we will not hold up the peer review process if you are unable.

Reviewers' comments:

Reviewer's Responses to Questions

**Comments to the Author**

1. Is the manuscript technically sound, and do the data support the conclusions?

Reviewer #1: Yes

Reviewer #2: Yes

2. Has the statistical analysis been performed appropriately and rigorously? 

Reviewer #1: Yes

Reviewer #2: Yes

3. Have the authors made all data underlying the findings in their manuscript fully available?

Reviewer #1: Yes

Reviewer #2: Yes

4. Is the manuscript presented in an intelligible fashion and written in standard English?

Reviewer #1: Yes

Reviewer #2: Yes

5. Review Comments to the Author

Reviewer #1: The authors have worked hard on data collection and writing the manuscript. It is concise and easy to read. The objectives are clear, and the paper is well-structured, well-written and has the potential to be published in this journal. However, the reviewer report highlights some minor translation or typo errors. Once revised, it will further enhance the manuscript and be more precise for the readers. Overall, reviewer recommend the paper for publication.

1. L46: Uncertainty is written twice. Correct it.

2. L133: Remove quotation mark and correct the phrase as uses and used are stated simultaneously (L213 as well).

3. L158, 165: Numbers are overlapping. Try another format.

4. L293-294: There are some minor translation errors, digitalization is mentioned twice. Same correction at L430.

5. L304: Remove ‘initiative was implemented’ since it is already expressed in L303.

6. L311, 319: Grassroot party building leadership, correct it.

Reviewer #2: The study investigates the formation of community resilience in the face of public health emergencies and identifies several typical patterns of community resilience based on the Chinese context. This aligns with the broader community governance challenges faced globally in recent years. The conclusions drawn are also somewhat in line with reality and have the potential to provide valuable insights for addressing real-world issues. However, there are still elements of the manuscript that need to be significantly improved to make a contribution.

1.The authors should consider incorporating terms like "digitalization" or "smart governance" in the title, as this is key to defining the research framework. This would also help readers better understand the study's context.

2.It is recommended that the authors clearly outline the research problem, conclusions, and innovations in the Introduction section.

3.As the core concept of the paper, the authors should provide a detailed definition of "community resilience." This should include a comparison of community resilience in the context of public health emergencies with its general definition, highlighting both similarities and differences. Additionally, the authors should discuss how their conclusions offer innovative insights compared to existing research on community resilience. This will further emphasize the value of exploring community resilience specifically in the context of public health emergencies.

4.A critical issue to address is how this research might provide insights applicable to other countries and regions beyond China.

5.The choice of five condition variables needs a detailed theoretical or practical justification. Additionally, the authors need to ensure clear differentiation between these variables. For instance, the manuscript uses “the identity of the team leader (provincial party secretary = 4, governor = 3, vice governor = 2, other = 1)” to measure Policy Support. However, in the Chinese context, this variable may also reflect aspects of Leadership and Party-building to some extant. Therefore, the authors should ensure the independence of the variables.

6.Clarification is needed on how variables like “Digitization of organizations” are measured. The explanation provided is insufficient: “The degree of digitization in organizations is measured through the digitalization of party and government institutions, as assessed by the “2021 China Digital Government Construction White Paper.” At what level of party and government is it based, and how is this ‘digitalisation’ measured? The authors need to address these questions in more detail. Similarly, the five indicators used to measure “community resilience” should be exemplified to aid reader understanding.

7.The Discussion section lacks content on the theoretical contributions of the study.

8.The Policy Insights section should provide more detailed recommendations on policy or practice. Current suggestions are somewhat broad and lack actionable guidance for practitioners.

9.The manuscript suffers from numerous language issues. For example: “Community resilience refers to a community's ability of a community to learn from each crisis...”; “The study proposes a 'qualitative' and 'quantitative' approach based on a complex dynamic perspective of histories."”; “The study uses the fsQCA method is used to analyze 31 provinces and cities in mainland China...”; “In this paper, we use data from 31 provincial-level regions in mainland China are selected as the sample.”; “we use the fsQCA method is used to study the factors that...”; “Digital Government The degree of construction of”; “grassroots party-building leadership of grassroots party building.”

10.There are numerous issues with the translation of references. For instance, "管理世界" should be translated as "Journal of Management World," not “Managing the World,” and "Theoretical Investigation" was incorrectly translated as “LILUN TANTAO.” The authors should carefully correct these errors.

11.The clarity of the Figures in the manuscript should be improved.

6. PLOS authors have the option to publish the peer review history of their article (what does this mean?). If published, this will include your full peer review and any attached files.

Reviewer #1: No

Reviewer #2: No

---

## [Author Response · Author response to Decision Letter 0]

16 Sep 2024

Dear Editorial Department of PLOS ONE:

Hello!

Based on the issues pointed out by the review experts and the proposed modification suggestions, the modification status of this article is briefly described as follows:

Question 1:Please ensure that your manuscript meets PLOS ONE's style requirements, including those for file naming. 

Modification 1: Modifications have been made as required.

Question 2:Please state what role the funders took in the study. If the funders had no role, please state: "The funders had no role in study design, data collection and analysis, decision to publish, or preparation of the manuscript."

Modification 2: Explanation of the role of sponsors has been added.

Question3:Please note that your Data Availability Statement is currently missing the repository name and/or the DOI/accession number of each dataset OR a direct link to access each database.

Modification 3: Add a database download link in the cover letter.

Question4：L46: Uncertainty is written twice. Correct it.

Modification4：Correct Uncertainty.

Question5：L133: Remove quotation mark and correct the phrase as uses and used are stated simultaneously (L213 as well).

Revision 5: The quotation marks have been removed and the phrase has been corrected.

Question6：L158, 165: Numbers are overlapping. Try another format.

Revision 6: The issue of overlapping numbers has been resolved.

Question7：L293-294: There are some minor translation errors, digitalization is mentioned twice. Same correction at L430.

Revision 7: Translation errors have been corrected.

Question8：L304: Remove‘initiative was implemented’ since it is already expressed in L303.

Revision 8：already Remove‘initiative was implemented’

Question9：L311,319: Grassroot party building leadership, correct it.

Revision 9: The statement made by grassroots party building leaders has been corrected.

Question10：The authors should consider incorporating terms like "digitalization" or "smart governance" in the title, as this is key to defining the research framework. This would also help readers better understand the study's context.

Revision 10: The term "digitalization" has been added to the title.

Question11：It is recommended that the authors clearly outline the research problem, conclusions, and innovations in the Introduction section.

Revision 11: The research question, conclusion, and innovation have been clearly outlined in the introduction section.

Question12：As the core concept of the paper, the authors should provide a detailed definition of "community resilience." This should include a comparison of community resilience in the context of public health emergencies with its general definition, highlighting both similarities and differences. Additionally, the authors should discuss how their conclusions offer innovative insights compared to existing research on community resilience. This will further emphasize the value of exploring community resilience specifically in the context of public health emergencies.

Revision 12: The community resilience in the context of sudden public health emergencies has been compared with its general definition in the introduction, highlighting the similarities and differences.

Question13：A critical issue to address is how this research might provide insights applicable to other countries and regions beyond China.

Revision 13: The discussion section in the text has already explained how the research can be applied to insights from countries and regions outside of China.

Question14：The choice of five condition variables needs a detailed theoretical or practical justification. Additionally, the authors need to ensure clear differentiation between these variables. For instance, the manuscript uses “the identity of the team leader (provincial party secretary = 4, governor = 3, vice governor = 2, other = 1)” to measure Policy Support. However, in the Chinese context, this variable may also reflect aspects of Leadership and Party-building to some extant. Therefore, the authors should ensure the independence of the variables.

Revise 14: Make indicator adjustments to ensure the independence of variables.

Question15：Clarification is needed on how variables like “Digitization of organizations” are measured. The explanation provided is insufficient: “The degree of digitization in organizations is measured through the digitalization of party and government institutions, as assessed by the “2021 China Digital Government Construction White Paper.” At what level of party and government is it based, and how is this ‘digitalisation’ measured? The authors need to address these questions in more detail. Similarly, the five indicators used to measure “community resilience” should be exemplified to aid reader understanding.

Revision 15: The selection criteria for each indicator have been explained in detail.

Question16：The Discussion section lacks content on the theoretical contributions of the study.

Revision 16: Content regarding the theoretical contribution of this study has been added to the discussion section.

Question17：The Policy Insights section should provide more detailed recommendations on policy or practice. Current suggestions are somewhat broad and lack actionable guidance for practitioners.

Revision 17: More detailed recommendations on policies or practices have been provided in the Policy Implications section.

Question18：The manuscript suffers from numerous language issues. For example: “Community resilience refers to a community's ability of a community to learn from each crisis...”; “The study proposes a 'qualitative' and 'quantitative' approach based on a complex dynamic perspective of histories."”; “The study uses the fsQCA method is used to analyze 31 provinces and cities in mainland China...”; “In this paper, we use data from 31 provincial-level regions in mainland China are selected as the sample.”; “we use the fsQCA method is used to study the factors that...”; “Digital Government The degree of construction of”; “grassroots party-building leadership of grassroots party building.”

Revision 18: Language issues in the manuscript have been corrected.

Question19：There are numerous issues with the translation of references. For instance, "管理世界" should be translated as "Journal of Management World," not “Managing the World,” and "Theoretical Investigation" was incorrectly translated as “LILUN TANTAO.” The authors should carefully correct these errors.

Revision 19: Translation issues with references have been corrected.

Question20：The clarity of the Figures in the manuscript should be improved.

Revision 20: The clarity of numbers in the manuscript has been improved.

Thank you for pointing out the issues and providing valuable revision suggestions. The research team has re proofread and checked the paper according to the editorial department's comments, and the revised parts have been highlighted in red. If there are any further modifications, please let me know. Thank you!

 Li Liqing , Li Zihan , Ding Haifeng and Gao Meng

September 14, 2024

---

## [Decision Letter · Decision Letter 1]

19 Nov 2024

PONE-D-24-10967R1How can digitalisation be used to develop community resilience in public health emergencies?: A qualitative comparative analysis from ChinaPLOS ONE

Dear Dr. li,

Thank you for submitting your manuscript to PLOS ONE. After careful consideration, we feel that it has merit but does not fully meet PLOS ONE’s publication criteria as it currently stands. Therefore, we invite you to submit a revised version of the manuscript that addresses the points raised during the review process.

We look forward to receiving your revised manuscript.

Kind regards,

Shenghua Xie

Academic Editor

PLOS ONE

Journal Requirements:

Reviewers' comments:

Reviewer's Responses to Questions

**Comments to the Author**

1. If the authors have adequately addressed your comments raised in a previous round of review and you feel that this manuscript is now acceptable for publication, you may indicate that here to bypass the “Comments to the Author” section, enter your conflict of interest statement in the “Confidential to Editor” section, and submit your "Accept" recommendation.

Reviewer #1: (No Response)

Reviewer #2: (No Response)

2. Is the manuscript technically sound, and do the data support the conclusions?

Reviewer #1: Yes

Reviewer #2: Yes

3. Has the statistical analysis been performed appropriately and rigorously? 

Reviewer #1: Yes

Reviewer #2: Yes

4. Have the authors made all data underlying the findings in their manuscript fully available?

Reviewer #1: Yes

Reviewer #2: Yes

5. Is the manuscript presented in an intelligible fashion and written in standard English?

Reviewer #1: No

Reviewer #2: Yes

6. Review Comments to the Author

Reviewer #1: (No Response)

Reviewer #2: I appreciate the authors’ effective improvements and their appropriate responses to some of my concerns in this round of revisions, which is commendable. However, to enhance the rigor of the manuscript and increase its chances of acceptance, the authors should address the following points:

1.Given that PLOS ONE is an international journal aimed at a global audience, I recommend that the authors remove certain culturally specific background elements related to China, such as references to specific government meetings and speeches by national leaders, as well as particular phrases like “modernizing the national governance system and capacity.”

2.Please review the clarity of the text, such as the phrasing in line 139: “of grassroots governance [19].,”

3.I suggest relocating the literature review from the “Introduction” section to a later chapter to address the redundancy in the first section.

4.The authors should also carefully verify the accuracy of the translations of cited Chinese literature, as I pointed out in the first round of feedback. For instance, the correct translation of “清华管理评论” is “Tsinghua Business Review,” not “Tsinghua Management Review.” I encourage the authors to ensure that each reference is accurately translated.

7. PLOS authors have the option to publish the peer review history of their article (what does this mean?). If published, this will include your full peer review and any attached files.

Reviewer #1: No

Reviewer #2: No

---

## [Author Response · Author response to Decision Letter 1]

28 Nov 2024

Response letter

Dear Editor, 

We would like to thank you and all reviewers for your insightful comments concerning the revisions of our manuscript entitled “How can digitalization be used to develop community resilience in public health emergencies?: A qualitative comparative analysis from Chinas”. We have carefully reviewed the reviewers’ comments, and we have addressed all their questions/critiques/concerns point-by-point as below. Mark all changes in tracked so that they may be easily identified. We believe that our manuscript has been significantly improved, and we hope it is now suitable for publication.

If you have any problems with our manuscript, please do not hesitate to contact us.

Review Report

The revised manuscript titled "How can digitalisation be used to develop community resilience in public health emergencies?: A qualitative comparative analysis from China" presents a well-structured and scientifically sound study. The research objectives are clear, the methodology is appropriate, and the results are significant and well-interpreted. The study contributes valuable insights to the field and meets the standards of this journal. However, there are numerous grammatical mistakes, translation and typo errors throughout the manuscript, making it difficult to recommend it in its current form for publication. 

1.There are still numerous grammatical and typo errors. Line 108 typo error (“), Line 139 (,), and similar issues can be seen throughout the manuscript. Please review thoroughly and correct such errors.

Response: Thank you for your feedback. We have carefully reviewed the manuscript and corrected all grammatical, typographical, and punctuation errors, including those in Line 108 and Line 139.

2.Figure 1 is missing. It does not show in this manuscript.

Response: Thank you for pointing this out. Figure 1 has been uploaded separately in the system according to the journal's submission requirements and was not included in the main text.

3.I suggest it is best to revise all phrases related to data sample areas as 31 provincial-level regions or 31 provinces or 31 cities to avoid confusions.

Response: Thank you for your suggestion. We have revised the manuscript to consistently use the term "31 provincial-level regions in mainland China" to avoid confusion.

4.Line 161 states configuration theory as theoretical framework for "holistic intelligent governance - community resilience formation", while Line 182-184 states holistic governance theory and digital governance theory as theoretical framework for “holistic wisdom governance-community resilience formation”. Clarify the theoretical framework and use one term “wisdom/intelligent” throughout the manuscript for better readability. 

Response: Thank you for your valuable comments. We have clarified the theoretical framework and unified the terminology throughout the manuscript. "Holistic intelligent governance" is now consistently used instead of varying terms.

5.There are typo errors in references sections. Add space between words, check reference 4,8,9,10… 

Response: Thank you for pointing this out. We have thoroughly reviewed the references section and corrected the typos, ensuring proper spacing between words and fixing references 4, 8, 9, and 10.

Reviewer #2: I appreciate the authors’ effective improvements and their appropriate responses to some of my concerns in this round of revisions, which is commendable. However, to enhance the rigor of the manuscript and increase its chances of acceptance, the authors should address the following points:

1.Given that PLOS ONE is an international journal aimed at a global audience, I recommend that the authors remove certain culturally specific background elements related to China, such as references to specific government meetings and speeches by national leaders, as well as particular phrases like “modernizing the national governance system and capacity.”

Response: Thank you very for your suggestions. This study mainly employs the fuzzy-set qualitative comparative analysis (fsQCA) method to establish a configuration research framework for the promotion of community resilience through holistic intelligent governance in response to public health emergencies. Using 31 provincial-level regions in mainland China as the research sample, we explore the configuration patterns and development paths of holistic intelligent governance that foster community resilience, all within the context of Chinese scenarios and issues. We sincerely appreciate your advice, and in our future research, we will broaden the scope of our study to include a global perspective.

2.Please review the clarity of the text, such as the phrasing in line 139: “of grassroots governance [19].,”

Response: Thank you for your feedback. The phrasing in line 139 has been revised for clarity.

3.I suggest relocating the literature review from the “Introduction” section to a later chapter to address the redundancy in the first section.

Response: Thank you for your suggestion. The introduction section was indeed lengthy, but it contains a description of the research background, which helps readers understand the context, continuity, and relevance of the study. It also ensures that the scientific rigor and feasibility of the research are clear to the readers. Additionally, the journal's submission guidelines require the introduction to be placed at the beginning. We appreciate your input, and in order to avoid excessive length, we have made some reductions to the introduction. Once again, thank you for your valuable feedback.

4.The authors should also carefully verify the accuracy of the translations of cited Chinese literature, as I pointed out in the first round of feedback. For instance, the correct translation of “清华管理评论” is “Tsinghua Business Review,” not “Tsinghua Management Review.” I encourage the authors to ensure that each reference is accurately translated.

Response: Thank you for your valuable feedback. We have carefully reviewed the translations of the cited Chinese literature and have made the necessary corrections, including updating “清华管理评论” to "Tsinghua Business Review." We will ensure that all references are accurately translated.

---

## [Editor Report · Decision Letter 2]

1 Dec 2024

How can digitalisation be used to develop community resilience in public health emergencies?: A qualitative comparative analysis from China

PONE-D-24-10967R2

Dear Dr. Li<!--EndFragment,

We’re pleased to inform you that your manuscript has been judged scientifically suitable for publication and will be formally accepted for publication once it meets all outstanding technical requirements.

Kind regards,

Shenghua Xie

Academic Editor

PLOS ONE
---

## [Editor Report · Acceptance letter]

10 Dec 2024

PONE-D-24-10967R2 

PLOS ONE

Dear Dr. Li, 

I'm pleased to inform you that your manuscript has been deemed suitable for publication in PLOS ONE. Congratulations! Your manuscript is now being handed over to our production team.

Kind regards, 

on behalf of

Dr. Shenghua Xie 

Academic Editor

PLOS ONE